# Development and Characterization of an In Vitro Cell-Based Assay to Predict Potency of mRNA–LNP-Based Vaccines

**DOI:** 10.3390/vaccines11071224

**Published:** 2023-07-10

**Authors:** Nisarg Patel, Zach Davis, Carl Hofmann, Josef Vlasak, John W. Loughney, Pete DePhillips, Malini Mukherjee

**Affiliations:** Analytical Research & Development, Merck & Co., Inc., 770 Sumneytown Pike, West Point, PA 19486, USA; npatel886@live.com (N.P.); zpdavis93@gmail.com (Z.D.); carl.hofmann@merck.com (C.H.); josef_vlasak@merck.com (J.V.); john_loughney@merck.com (J.W.L.); pdsld@comcast.net (P.D.)

**Keywords:** mRNA, LNP, vaccines, potency, potency assays, bioassays

## Abstract

Messenger RNA (mRNA) vaccines have emerged as a flexible platform for vaccine development. The evolution of lipid nanoparticles as effective delivery vehicles for modified mRNA encoding vaccine antigens was demonstrated by the response to the COVID-19 pandemic. The ability to rapidly develop effective SARS-CoV-2 vaccines from the spike protein genome, and to then manufacture multibillions of doses per year was an extraordinary achievement and a vaccine milestone. Further development and application of this platform for additional pathogens is clearly of interest. This comes with the associated need for new analytical tools that can accurately predict the performance of these mRNA vaccine candidates and tie them to an immune response expected in humans. Described here is the development and characterization of an imaging based in vitro assay able to quantitate transgene protein expression efficiency, with utility to measure lipid nanoparticles (LNP)-encapsulated mRNA vaccine potency, efficacy, and stability. Multiple biologically relevant adherent cell lines were screened to identify a suitable cell substrate capable of providing a wide dose–response curve and dynamic range. Biologically relevant assay attributes were examined and optimized, including cell monolayer morphology, antigen expression kinetics, and assay sensitivity to LNP properties, such as polyethylene glycol-lipid (or PEG–lipid) composition, mRNA mass, and LNP size. Collectively, this study presents a strategy to quickly optimize and develop a robust cell-based potency assay for the development of future mRNA-based vaccines.

## 1. Introduction

Messenger RNA (mRNA) vaccines represent an innovative and emerging platform technology for vaccine development. Relative to common vaccine modalities, often attenuated or inactivated pathogens or subunit vaccines, mRNA vaccines offer the potential for a flexible platform, with rapid, scalable and cell-free production processes, and the ability to accommodate almost any immunogen, and potentially multiple immunogens [1,2]. The speed and utility of this approach was exemplified by the remarkable response to the COVID-19 pandemic, which saw the development, clinical evaluation, and regulatory authorization of two mRNA vaccines within a year of the publication of the SARS-CoV-2 virus spike genome [3].

LNPs are frequently used for the delivery of mRNA, building upon advances and progress made since the development of LNP delivery systems for the intravenous (IV) administration of small interfering RNA (siRNA). In an earlier work, the composition and biophysical properties of LNP delivery vehicles were explored and optimized, to develop stable particles that can protect RNA from digestion by nucleases, and promote efficient cellular uptake and subsequent RNA escape from the endosome to the cytoplasm. The resulting compositional framework, with a formulation consisting of an ionizable lipid, cholesterol, a PEGylated lipid, and a helper lipid, often 1,2-distearoyl-*sn*-glycero-3-phosphocholine (DSPC) [4,5], became an important vehicle for demonstrating the immunogenicity of mRNA vaccines in animal and clinical studies. Ionizable amino lipids with pKa values between 6 and 7 were found to have a multifunctional role in siRNA potency, enabled by their pH sensitivity [6]. Under acidic conditions, positively charged amino groups facilitate ionic interactions with siRNA during particle assembly, and the release of the siRNA from acidic endosomes after LNP uptake, while remaining neutral at physiological pH. PEGylated lipid in LNP formulations provides a “steric stabilization” effect through the formation of a hydrophilic layer on the particle surface, preventing particle aggregation, and reducing uptake by macrophages of the mononuclear phagocyte system (MPS) [7,8].

The cellular uptake and delivery of siRNA or mRNA LNPs is a complex, multi-step process, beginning with the attachment of the LNP to the cell surface, cell entry by endocytosis, release of the RNA from the endosome to cytosol, and binding of the RNA to the RNA-induced silencing complex (RISC) for gene silencing, or to a ribosome for protein expression. The cell recognition and uptake process begins when LNPs come in contact with biological fluids, which rapidly results in the covering of the LNP surface with a corona of biomolecules, including immunoglobulins, lipoproteins and coagulation factors [8,9]. Constituents of this corona act as ligands for specific cell surface receptors, initiating receptor-mediated endocytosis, and subsequent LNP internalization [10]. Apolipoprotein E (ApoE) acts as the key endogenous ligand in this role, able to bind to ionizable LNPs enabling receptor-mediated endocytosis by low-density lipoprotein receptors (LDL-R) [11,12], which are involved in cholesterol utilization and lipid homeostasis [13], and therefore widely distributed on all nucleated cells. The exchange of ApoE from chylomicrons and high-density lipoproteins results in the coating of the LNP surface with a relatively high concentration of ApoE ligands, resulting in high avidity for LDL-R. Shielding of the lipid surface by a PEG layer modulates the formation of the protein corona, so that increasing PEGylated lipid composition decreases opsonization and corona formation, reducing cellular uptake of LNPs and significantly increasing in vivo half-life [8,14,15,16].

Receptor-mediated endocytosis is both size dependent and size selective, with the cell entry process induced and energetically driven by the specific interactions between the cell surface receptors and their cognate ligands on the nanoparticle surface [17]. Endocytosis kinetics are therefore dependent on sufficient density of both ligand and receptor. The fundamental relationship between nanoparticle size and uptake kinetics is described in a recent model [18], in which particle size and ligand density were found to be the key parameters which defined a narrow size window of 27–30 nm in which cell uptake is optimal and rapid. Within this size range, ligand and receptor density were not limiting. As nanoparticle size is decreased below this optimum, endocytosis progressively slows due to insufficient ligand density. At nanoparticle sizes larger than optimal, many more ligand–receptor interactions are required to complete the membrane wrapping process, resulting in the progressive slowing down of endocytosis as size increases, due to the depletion of the available receptors within diffusible distance of the particle.

Additional variables inherent to both the nanoparticles (shape, and surface chemistry) and cell surfaces (elasticity, receptor concentration and diffusivity) makes the nanoparticle–cell interface highly heterogenous [18]. So, the cell uptake kinetics and optimal particle size window may vary significantly with different cell types and nanoparticle composition. 

Because the size and physicochemical properties of LNPs can significantly affect both stability and in vivo delivery, more recent studies have continued the development of this formulation for mRNA vaccines. This reflects the need for vaccine specific intramuscular (IM) administration, the corresponding desire to optimize immunogenicity, and to identify ionizable lipids with better injection tolerability and fewer reactogenic events [19,20].

In order to fulfil these aims, there is a need for well-defined and well-characterized assays to quantitate transfection efficiencies and levels of protein antigen expression in appropriate cell types, to facilitate development of mRNA vaccine candidates, and ensuring lot-to-lot consistency during manufacture. 

Cell-based assays can examine functional LNP attributes needed for cell uptake and endosomal release, in addition to mRNA translatability, and would complement cell-free in vitro translation (IVT) assays in measuring mRNA LNP potency, efficacy, and stability [21]. 

Respiratory syncytial virus (RSV) affects millions of infants and young children as well as the elderly population globally with respiratory illnesses of varying severity leading to a high rate of outpatient visits and even hospitalizations [22,23]. Currently, there is no treatment for RSV other than supportive care and a vaccine against RSV infection is a high unmet need. The pre-fusion form of the fusion (F) protein in RSV is responsible for cellular entry and has the potential to elicit high RSV neutralizing antibody titers with a correlation to reduced disease severity. RSV-F has thus been an attractive target for vaccine development across several platforms [24]. The most recent approval of Arexvy by GlaxoSmithKline as the first RSV vaccine for adults older than 60 marks a big milestone in the protection of the elderly population against this disease. However, approval of vaccination for young children is still pending.

Described here is the step-by-step development and characterization of a high-throughput in vitro assay to predict potency of our mRNA–LNP based vaccine candidate encoding for RSV pre-fusion protein. We started the assay development with the selection of the most appropriate cell substrate based on the criteria of providing a wide dose–response and dynamic range as well as the ability to discriminate transfection and expression differences in LNP properties of biological relevance to potency and stability, such as size and mRNA content. This assay uses immunostaining to quantitate the number of cells positively expressing antigen protein, and the amount of antigen protein expressed per cell, providing the flexibility to quantitate vaccine relative potency or vaccine efficacy [25].

## 2. Materials and Methods

### 2.1. Cell Culture

HepG2 (ATCC, Washington, DC, USA, HB-8065), Vero (MSD, Rahway, NJ, USA, passage 124), HeLa (ATCC, CCL-2), Hep-2 (Millipore-Sigma, Burlington, MA, USA, 86030501), A549 (ATCC, CCL-185) and Caco-2 (ATCC, HTB-37), were obtained and maintained on a CompacT SelecT automated cell culture system (Sartorius). All cells were cultured in tissue culture-treated, barcoded T175 flasks (Corning, Corning, NY, USA, 431606) with cell specific mediums supplemented with 10 percent fetal bovine serum (FBS; ATCC, 30-2020) and 1 percent penicillin–streptomycin (Pen/Strep; Gibco, Grand Island, NY, USA, 15140-122). Vial thaw was performed in a 37 °C water bath, and upon thawing, cells were diluted 1:10, pelleted at 100 g for 6 min and resuspended in fresh culture medium after removing the diluted cryopreservation medium. The diluted cells were expanded and maintained on T175 flasks containing 70 mL of growth medium at various cell densities for 5–7 days (cell line dependent) in T175 flasks at 37 °C, 5% CO_2_, and ≥90% rH on the CompacT SelecT system.

### 2.2. RSV-F Protein Expression Assay

HepG2, A549, and Vero cells were harvested from T175 flasks in their respective culture media with the FBS content being kept constant at 2 percent for cell plating. HepG2 cells were seeded at a density of 7.5 × 10^4^ cells per well in 100 µL of plant media comprising EMEM (Corning, 10-009-CV), 2% FBS, 1% Pen/Strep on Corning^®^ 96-well tissue culture-treated microplates (Corning, 3904BC) or at a seeding density of 3.0 × 10^4^ cells per well in 100µL of plant media in 96-well BioCoat™ collagen I-coated microplates (Corning, 356649). A549 and Vero cells were seeded at a density of 3.0 × 10^4^ cells per well in 96-well tissue culture (Corning, 3904BC)-treated microplates. The plated cells were allowed to attach overnight at 37 °C, 5% CO_2_, and >90% rH. On the following day, reference standard, positive control and test articles of investigational vaccine stocks of mRNA sequences encased in cationic lipid nanoparticles (mRNA–LNP, and MSD) were pre-diluted to 2.5 μg/mL RNA in OptiMEM Reduced Serum Media (Gibco, 31985-070), then plated and titrated 1.667-fold in Corning^®^ Ultra-Low Attachment 96-well microplates (Corning, 7007) using the reduced serum medium. Following titration, 40 μL of the diluted material was transferred to the pre-seeded cell plates that underwent a medium exchange in plant medium. Transfection of cells was allowed by co-incubation with mRNA–LNP batches in presence of media described above at 37 °C, 5% CO_2_, and 90% RH for 16 ± 2 h followed by cell fixation with 3.7% formaldehyde (Millipore-Sigma, F1635) in phosphate-buffered saline (PBS). Post-fixation, plates were stored in PBS prior to immunostaining. 

Fixed cells were permeabilized with 0.5% Triton X-100 (MSD) for 20 min, decanted, then promptly blocked with 1% bovine serum albumin (BSA; Teknova, Hollister, CA, USA, Cat# P1391) for 30 min. Cells were then fluorescently stained with RSV-F protein specific monoclonal antibody (mAb; MSD), rinsed twice with 0.1% Tween-20 wash buffer (Teknova, Cat# P0238) for 5 min, followed by goat anti-species IgG Alexa Fluor^®^ 488 fluorophore conjugate (Invitrogen, Waltham, MA, USA, A-11013) for identification of cells expressing the protein of interest. Additionally, cells were co-stained with Hoechst 33,342 nucleic acid dsDNA stain (Invitrogen, H3570) to label the entire cell population independent of the mAb-targeted protein. Cells were then rinsed twice in 0.1% Tween-20 wash buffer for 5 min, followed by buffer exchange in PBS for storage until imaging. 

Immunostained cell plates were finally imaged on the BioTek^®^ Cytation™ 3 or Cytation™ 5 automated fluorescence microscopy imager using BioTek^®^ Gen5™ software (version: 3.14.03). Fluorescence intensity and size criteria were applied for image segmentation using Gen5™ to quantitate cellular events. The segmentation results in a count of viral analyte expressing cells and a count of the total cell population in a given well. The ratio of analyte-expressing cells to total cells was performed to calculate the % transfected cells at each mRNA dose; this dose–response relationship is then plotted, and a four-parameter logistics (4-PL) regression is applied to calculate the potency of each test article relative to the reference standard.

### 2.3. ApoE Supplementation

HepG2, Vero and A549 cells were cultured, plated at 3.0 × 10^4^ cells per well and allowed to attach overnight into 96-well microplates as stated before. Apolipoprotein E purified from human plasma (Millipore-Sigma SRP6303-50 ug) was introduced into the assay by supplementation in the OptiMEM dilution media at 4 µg/mL and 20 µg/mL during the pre-dilution and titration steps. For control condition, mRNA–LNP vaccine stock was diluted in dilution medium without supplementation. Each cell line was treated with two replicate titrations of mRNA–LNP vaccine stock with ApoE supplementation at 4 µg/mL ApoE and 20 µg/mL, or with non-supplemented control. Transfection and fluorescent immunostaining were performed as per before. Dose response relationship was established using 4-PL logistics regression and relative potency was calculated back to the non-supplemented control.

### 2.4. LDL Receptor Comparison

HepG2 cells were seeded at 7.5 × 10^4^ cells per well while Vero and A549 cells were seeded at 3.0 × 10^4^ cells per well in 96-well tissue culture microplates in cell-specific plant medium. The cells were allowed to attach overnight at 37 °C, 5% CO_2_, and >90% rH. Cells were transfected in an additional experiment the following day and all cells were incubated an additional 16 ± 2 h for transfection, as previously mentioned. Untreated wells of each cell line were permeabilized and blocked as before and then fluorescently tagged with recombinant anti-LDLR mAb, (Abcam, Cambridge, UK, AB52818) followed by goat anti-rabbit IgG Alexa Fluor 488 (Invitrogen, A11034). Cells were again co-stained with a nuclear stain for cellular identification and then imaged using BioTek Cytation3 imager.

### 2.5. HepG2 Cell Layer Optimization

HepG2 cells were cultured and harvested in plant medium as before. The cells were then plated in tissue culture microplates at a seeding density of 7.5 × 10^4^ cells per well and in collagen I-coated microplates at seeding densities of 7.5 × 10^4^, 5.0 × 10^4^, and 2.5 × 10^4^ cells per well for an initial comparison screen of cell layer. Cells were allowed to attach overnight as before and then imaged using a phase contrast microscope. In a follow-up experiment, HepG2 cells were seeded at densities of 2.5 × 10^4^, 3.0 × 10^4^, and 3.5 × 10^5^ cells per well, followed by transfection, immunostaining and microscopy imaging being carried out as before for a further optimization of cell layer.

### 2.6. Density Gradient Centrifugation

Density gradient centrifugation (DGC) was run under differential velocity mode. Linear density gradient was prepared using a gradient maker. The gradient consisted of 2–10% sucrose and an opposite gradient of NaCl (120–0 mM) to maintain identical osmolality throughout the gradient (exposing LNPs to lower osmolality may result in size increase due to water uptake). The base buffer was 20 mM Tris pH 7.4. Briefly, 11 mL gradient was prepared and 0.5 mL of sample (approximately 20 mg/mL total lipid concentration) in a buffer containing 14% sucrose was layered on the bottom of the gradient with a gel-loading tip. Due to the high lipid content, the LNPs are buoyant and during centrifugation travel from the bottom of the tube to the top. The tubes were spun at 20,000 rpm for 40 min in SW40 Ti rotor. After centrifugation, about 1 mL fractions of mRNA–LNP particles separated based on their size and density were removed from the top using a gradient fractionation device (Biocomp Instruments, Fredericton, Canada).

## 3. Results and Discussion

### 3.1. Initial Cell Line Screening Revealed Significant Differences in RSV-F Protein Expression

Six established adherent cell lines, with properties as described in Table 1, were selected for an initial screening to compare LNP transfection efficiency and RSV-F protein expression. These cell lines are commercially available, with five of them, A549, HepG2, HEp-2, Caco and HeLa having demonstrated lipid nanoparticle endocytosis in previous studies [26,27,28,29,30], and Vero representing a cell line often used for large scale manufacturing of a variety of vaccines, with well-known cell culture and growth conditions. 

For this comparison, all cell lines, except HepG2, were seeded in 96-well plates at a density of 3.0 × 10^4^ cells per well in a 100 uL of planting media composed of corresponding basal media and 2% FBS. HepG2 cells required seeding at a density of 1.25 × 10^5^ cells per well to achieve confluence (>90%) within wells. LNPs were added to cells at multiple concentrations,16 h after addition cells were fixed and permeabilized. Nuclei staining to count the total number of cells/well, and immunostaining with anti-RSV-F protein mAb to identify and count the subpopulation of cells positively expressing RSV-F protein was carried out as described in materials and methods. The percentage of total cells positively expressing RSV-F protein is used as a measure of the efficiency of the combined transfection (endocytosis, and endosome escape) and translation processes, and is referred to here as RSV-F or protein expression efficiency.

RSV-F expression efficiency varied significantly between cell lines. HepG2 cells showed a sigmoidal dose–response curve, with RSV-F expression in ~90% of cells at the upper asymptote, as shown in Figure 1A,G, brown line. In contrast, A549 cells showed no detectable protein expression (Figure 1A,F, purple line), while RSV-F expression in Vero (Figure 1A,C, blue line), HeLa (Figure 1A,B, red line), HEp-2 (Figure 1A,D, green line) and CaCo-2 (Figure 1A,E, orange line) exhibited a pronounced hook effect, with expression efficiency increasing with dose until peaking at ~50% (HeLa) or ~60% (Vero) of cells, and then declining at higher doses, with very low RSV-F expression observed at the highest dose.

Rapid and efficient cell uptake of LNPs by receptor-mediated endocytosis requires adequate amounts of both the cell surface receptor, LDL-R, and the cognate ligand on the LNP, ApoE. Differences in LDL-R density on the cell surface for different cell lines, or differences in the concentration of ApoE in the media during transfection could each contribute to the significantly different protein expression efficiencies observed here. Although some of the cell lines used here are derived from tissues which express ApoE (liver, and kidney), the amount of endogenous ApoE produced and secreted by specific cell lines is variable, and not always known. For example, the human hepatoma HepG2 cells, and CaCo-2 intestinal epithelial cells are known to synthesize and secrete ApoE [31,32,33] but secreted ApoE was not detectable in HeLa cells [33]. Endogenous ApoE is present in the FBS added to the media, but at unknown concentration. To ensure that an inadequate concentration of ApoE is not limiting LNP uptake, comparisons of cell lines were also carried out with ApoE spiked into media at transfection, to a concentration of 4 ug/mL.

Vero, HeLa, and HEp-2 cells showed a pronounced hook effect in the absence of added ApoE; the addition of ApoE to a concentration of 4 ug/mL eliminated the hook effect, and enabled RSV-F expression in nearly all cells (91–99%), as shown in Figure 1B,D, dotted line. A459 and the Caco-2 cells, on the contrary, show limited enhancement of transfection upon ApoE supplementation (Figure 1E,F, dotted line), while the HepG2 cells did not show a significant change with addition of ApoE (Figure 1G, dotted line). The significantly improved RSV-F expression efficiency seen in some of these cell lines is indicative that LDL-R is present at sufficient concentration on the cell surface to support rapid LNP uptake by receptor-mediated endocytosis, but efficient protein expression in these cell lines was limited by insufficient ApoE and could not be achieved unless the media was supplemented with additional ApoE. HeLa cells are known to not secrete ApoE [33], and the similar hook effects seen in the absence of added ApoE with Vero and HEp-2 are consistent with limited production and secretion of ApoE by these cell lines, and therefore potentially reduced ability for endocytosis at higher LNP concentrations, when endogenous ApoE in FBS (at 2%) is also relatively low. Representative images of HepG2, Vero and A459 cells transfected with RSV-F mRNA containing LNP in the absence (Figure 1H, left panel) and presence of ApoE (Figure 1H, right panel) are shown here.

Finally, although ApoE addition enabled efficient protein expression in Vero, HeLa and HEp-2 cells, it also resulted in a significant increase in the curve hill-slope values [34], with values ranging from ~3.5 to 9 (Figure 1B–D, dotted line). The increased transfection seen in some cells in the presence of ApoE are indicative of an overall improvement in transfection efficiency, and are consistent with increased endocytosis, but additional factors may be involved. A recent study demonstrated that ApoE binding to mRNA–LNP resulted in a restructuring of the surface and core, leading to a measurable decrease in encapsulation, and allowing mRNA to escape from the LNP [35]. These insights provide an additional hypothesis for the unusually steep slopes observed here in the presence of high concentrations of ApoE, and further follow-up is necessary for a more mechanistic understanding. The steep slopes would not be desirable for a potency assay, as they represent a significant compression of the dose range of the assay. In addition, steeper slopes mean that normal variability in the measurement of the RNA concentration used to determine the assay dose would result in correspondingly larger changes in potency, thereby amplifying total assay variability. These cell lines would not be ideal substrates for a vaccine potency assay without further optimization and control of ApoE concentration to produce more typical assay slope values.

### 3.2. Optimization of the Morphological Characteristics of the HepG2 Cell Layer for an mRNA Potency Assay

The precision and accuracy of image-based analysis is enhanced by a uniform cell monolayer, with minimal cell stacking and three-dimensional growth, so that the segmentation and counting of objects (cell nuclei and expressed protein) occurs in a predictable and defined focal plane. In addition, the shielding of cells in the interior of clustered or stacked cells may limit access and exposure to test articles and media components.

HepG2 cells are known to cluster and grow in three-dimensions [31,36], and this was observed for HepG2 cells when plated on a hydrophilic tissue culture plates at a density sufficient to achieve a confluency of ~90%, 7.5 × 10^4^ cells/well, as shown in Figure 2A. 

Hepatocytes in vivo are supported by an extracellular matrix composed primarily of fibronectin and collagen [37] and when cultured on type 1 collagen are able to form a stable monolayer, assuming a more polygonal phenotype with less rounding. [38]. Type 1 collagen-coated plates have been used to culture hepatocytes, as they more closely simulate in vivo conditions, and were examined here as an alternative to hydrophilic cell culture plates, at three different cell seeding densities: 7.5, 5.0 and 2.5 × 10^4^ cells/well. 

At a seeding density of 7.5 × 10^4^ cells per well in both collagen and cell culture plates, cell clustering and 3D structures were observed in images (from the center sector) of wells, as shown in Figure 2A,B. These 3D structures were successively reduced in number as seeding densities were reduced to 5.0 and 2.5 × 10^4^ cells per well in collagen plates (Figure 2C,D), with the latter density providing an even monolayer coverage, with confluency > 90%. 

The lower cell densities on collagen plates provided a more uniform cell monolayer, with minimal 3D growth, and protein expression efficiency was compared across a lower range of cell densities, 2.5, 3.0 and 3.5 × 10^4^. At these densities, the image segmentation and analysis to identify and count nuclei and the subpopulation of transfected cells was found to be accurate and reproducible, with nearly overlapping curves, as shown in Figure 2E. A cell density of 3.0 × 10^4^ provided robust results in the center of this lower range, and was selected as the plating density for subsequent mRNA vaccine potency assays described in rest of this study.

RSV-F protein expression in HepG2 cells was compared at 4, 6, 8, and 16 h post transfection, as shown in Figure 2F. Protein expression occurred rapidly, with measurable protein in ~30% of the cells at 4 h. Increasing the transfection time to 6 or 8 h increased the expression efficiency to ~70% with a corresponding reduction in EC50 suggesting an increase in % of cells positively expressing RSV-F protein at a given mRNA–LNP dose (tabulated in Figure 2G) and hence, improved assay sensitivity. At 16 h, nearly complete transfection (>95%) of the cells was achieved. Although protein expression is measurable at shorter transfection times, complete transfection and expression is optimal for a potency assay, providing maximum assay sensitivity and dynamic range, and 16 h was selected as the optimum potency assay transfection time.

### 3.3. Assay Sensitivity to mRNA Mass per LNP

The antigen mRNA is susceptible to a number of degradation reactions, including hydrolysis and oxidation [39], and chemical modification through formation of lipid adducts [40]. Chemical degradation or modification of mRNA which impedes or prevents translation in vivo would be detrimental to vaccine potency and efficacy. The ability of an in vitro potency assay to discriminate changes in mRNA content, particularly time, temperature and pH mediated changes by known degradation pathways, provides a stability indication with utility in optimizing solution and storage conditions. Verification of the ability to measure a potency loss resulting from mRNA degradation is difficult without an authentic set of LNPs that vary in the extent of mRNA degradation, and which have the specific degradation products identified and quantified. Current analytical characterization tools are not capable of detecting and quantitating all such mRNA degradation events that could occur in LNPs in accelerated stability studies, particularly on a per LNP basis, so preparing a fully defined set of comparator samples is not possible.

In lieu of authentic degraded samples, a set of samples prepared with different N/P ratios, as determined by the amine (N) groups of lipids to the phosphates (P) on mRNA [41], was used to assess the ability of this assay to quantitatively detect differences in mRNA mass per particle as a surrogate for loss of mRNA, although the “loss” is artificially created by increasing the N/P ratio to decrease the amount of mRNA encapsulated per LNP, and not by chemical degradation or modification of the LNP mRNA. The sample set consisted of LNPs at the target N/P ratio of 6, and N/P ratios of 3, 9, and 12, which bracket the target, and contain 2X, 0.67X and 0.5X the mass of mRNA per LNP, respectively, relative to the target ratio.

The resulting assay data were graphed three ways: (1) as % cells positively expressing RSV-F protein vs. RNA mass (typical for a potency assay), in Figure 3A, (2) mean fluorescence per cell vs. RNA mass, in Figure 3B. (typical for an efficacy assay), and (3) as % cells positively expressing RSV-F protein vs. lipid mass, in Figure 3C.

In Figure 3A, the curves for the LNPs with different N/P ratios largely overlap, with only small differences between EC50 values, and therefore equivalent potencies. In this assay, the LNP dose is based on the measured RNA concentration, and so changes in RNA concentration with changes in the N/P ratio are compensated for by a proportional (but inverse) increase or decrease in the number of particles used for transfection at any given dose, so that mRNA mass remains equal.

The upper asymptotes in Figure 3A no longer converge, varying inversely with the N/P ratio, so that at the lowest N/P of 12, the asymptote plateau corresponds to ~80% of the cells transfected. This may be due to increasing saturation of the endocytosis machinery by the larger number of LNPs needed to achieve equivalent RNA mass as the N/P ratio is increased above six. If so, although transfection may be occurring at comparable rates for all of different N/P ratio particles, if there is an upper limit on transfections per cell based on saturation, the higher N/P ratio particles will deliver less total RNA to the cells, resulting in less protein expressed per cell. 

This is consistent with the data shown in Figure 3B, where mean fluorescence intensity (MFI) per cell is plotted on the *y*-axis. These curves are non-convergent at the higher doses, indicating significant differences in the amount of RSV-F protein expression per cell as N/P ratios change. Protein expression per cell is increased as higher RNA concentration per particle is achieved at lower N/P ratios. Although all of the N/P ratios showed equivalent potency (equivalent EC50), the difference in maximal protein expression at the high dose is indicative of significant efficacy differences, as more protein translation per cell is achieved with higher RNA content (lower N/P ratio).

The same data are graphed in Figure 3C, where the *x*-axis (dose) represents lipid mass. Because lipid mass is relatively consistent for individual LNPs, it is used as an approximation for particle number, so that this graph compares expression efficiency as a function of the number of LNP particles.

In this visualization of the data, the increase in EC50 as the N/P ratio increases is significant, increasing ~2 fold with each 2-fold increase in N/P ratio. The LNPs at different N/P ratio are clearly resolved, and increases in N/P ratios result in decreases in both potency and efficacy which are both quantifiable when graphed this way. 

As discussed above, particles with different N/P ratios have equivalent potencies when graphed by RNA dose because dosing by RNA mass allows compensation for the differences in RNA concentration per particle by adjusting the number of particles per dose. 

In the absence of authentic and well-characterized LNPs with degraded RNA of known quantity, comparing the N/P series provides an experimentally tractable approach to confirm a stability indication for this assay. The data in Figure 3 demonstrate the ability of the assay to detect potency differences arising from different masses of RNA per LNP particle, which is analogous (but not identical to) to loss of RNA resulting from chemical degradation or modification.

### 3.4. Assay Sensitivity to LNP Size and PEG–Lipid Composition

The relationship of LNP size to potency in HepG2 cells was characterized by separating LNP preparations into size fractions using density gradient centrifugation. Two samples prepared at a composition of 1% or 2% 1,2-Dimyristoyl-sn-glycero-3-methoxypolyethylene glycol (PEG-DMG) were compared. For the sample with 2% PEG–lipid, the T-mixing ratio was modified to increase particle size, producing a pair of samples with nearly equal mean diameter (101 and 98 nm) and relative potency (234 and 274%), despite the different PEG–lipid composition. The diameter of the LNPs in each fraction was determined via dynamic light scattering (DLS) and graphed against the relative assay potency values in Figure 4A. The values for the unfractionated samples are shown as open symbols. Each sample was separated into six LNP fractions with sizes ranging from 79 to 122 nm in diameter for the 1% PEG–lipid sample and 69 to 115 nm for the 2% PEG–lipid sample, shown as closed symbols. 

Although mean diameter and potency were equivalent for the samples of 1 and 2% PEG-DMG, fractionation by DGC showed differences in the distribution of LNP sizes and the potency–size relationship with potency values ranging from 128 to 260% for 1% sample and 55 to 358% for 2% sample, relative to the unfractionated material. For both samples, the potency values were the highest in the middle of the size distribution and decreased for smaller and larger particles. This difference was more significant for the 2% PEG–lipid sample, reaching a 6-fold difference between the peak potency for 106 nm fraction and the lowest potency for 69 nm fraction. This size-related potency distribution is generally consistent with receptor mediated endocytosis as the entry mechanism for LNP uptake, resulting in an energetically preferred size range where endocytosis occurs more rapidly [16]. When considering fractions of comparable size, the potency of 1% PEG-DMG LNPs is measurably and consistently lower than those of 2% composition, for all fractions with diameters > 80 nm, as shown in Figure 4A. This potency difference is related to other effects of the PEG-DMG composition, and not its effect on LNP size.

The LNP size sensitivity of HepG2 cells in this potency assay is well-suited for use with LNP–mRNA vaccines, where 100 nm +/− 30 nm is a common size range. Potency decreases at LNP sizes outside this range are readily quantifiable, as shown in Figure 4B, where LNP lots with overlapping size distributions, but different mean diameters, are compared. The relative potency of the larger LNP lot (D = 170 nm) is 58% of the smaller LNP lot (D = 84 nm), a significant difference which is easily discriminated by this assay.

PEG-DMG lipid composition was varied to prepare mRNA–LNPs at 0.75, 1.5, 3.0, and 4.5%. Transfection efficiency and LNP diameters for these compositions are shown in Figure 4C,D. As expected and consistent with previously demonstrated effect of PEG shielding [14,42], increasing PEG-DMG above 1.5%, to 3.0 and 4.5% resulted in increased PEG shielding of the LNP surface, reducing cell uptake by limiting corona formation and ApoE exchange. At a composition of 4.5% PEG-DMG, no protein expression was detectable at 16 h post-transfection. A significant reduction in transfection efficiency was seen for the 0.75% PEG LNP, indicated by the right shift of the curve in Figure 4C, as compared to 1.5% PEG. This decrease in transfection efficiency at 0.75% PEG would not be explained by PEG shielding.

Potency reduction shown in Figure 4C for 0.75% PEG LNPs could be related to increased LNP size. The impact of PEG–lipid concentration on LNP size was shown previously [43] for empty LNPs produced via microfluidic synthesis, where LNP size over the range of 20–100 nm was controlled by variation in the PEG–lipid content. Because the PEG–lipid is preferentially incorporated on the LNP exterior, it was found to provide stability to the LNP and limit further growth through additional lipid incorporation. The reduced potency seen for the 0.75% PEG-DMG LNPs is consistent with the reduced potencies observed for larger particle fractions obtained from density gradient fractionation (Figure 4A). Alternatively, potency reduction for the 0.75% PEG LNPs could be related to the stability of the LNPs in culture medium. Low PEG–lipid shielding has been implicated in a loss of the cationic lipid, which subsequently reduced the efficacy of endosomal escape [42]. In fact, the data in Figure 4A show that for particles of the same size, a potency difference between 1 and 2% PEG–lipid samples still exists and it is therefore related to other effects of the PEG–lipid than its impact on particle size. Because the efficacy of mRNA delivery to cytosol can be affected by multiple properties of the LNPs, such as size, lipid content, or others [18], understanding the contributions of these individual parameters would require carefully controlled experiments. Importantly, the potency assay described here shows sensitivity to these relevant variations of the LNPs. 

## 4. Discussion

This study represents the step-wise development and optimization of a cell-based potency assay for mRNA vaccines. The comparison of several cell lines that readily uptake LNPs and express mRNA antigens showed significant differences in their ability to provide measurement attributes desired for a quantitative potency assay, particularly the wide dynamic range enabled by full transfection. The utility of three widely used adherent cells lines—HeLa, Vero, and HEp-2—as a potency assay cell substrate could be improved by supplementation of ApoE to the media during transfection to eliminate the pronounced hook effect; however, the very high hill-slopes resulting from excess ApoE significantly compress the useful concentration range of the assay, thus making these cell lines less desirable for this purpose. HepG2 cells make and excrete ApoE, and were found to provide sufficient LDL-R and ApoE for full transfection and greater dynamic range, and hill-slopes consistent with a desired wide analyte concentration range. However, the tendency to clump and not form uniform monolayers, still makes the HepG2 cells difficult to handle, especially in a large scale commercial manufacturing setup. To further optimize this, the use of coated plates was explored. The use of collagen plates with HepG2 provides consistent monolayer coverage at confluency, an optimal morphology suitable for use with imaging systems, and automated object recognition and quantitation for high throughput applications. This mRNA–LNP vaccine potency assay is sensitive to changes in biologically relevant LNP properties, such as mRNA concentration, LNP size, and PEG levels, providing a useful method for mRNA vaccine development and manufacturing control. The relative potency of vaccine lots can be measured by comparison of protein antigen expression efficiency of test samples to a reference material, while efficacy can be measured by the quantitation of antigen expression per cell.

When transfected in media containing 2% FBS, but no additional ApoE, A549 and CaCo-2 cell lines expressed little to no detectable RSV-F-protein, even at the highest doses. Addition of ApoE to the media significantly increased protein expression efficiency, to ~15% of cells for Caco-2 and ~30% of cells for A549, as shown in Figure 1E,F, but still remained significantly lower than the other cell lines tested. Adding ApoE ensured sufficient ligand density on the LNP surface, so the limited protein expression may indicate additional constraints on endocytosis, potentially in the number and distribution of clatherin pits and/or LDL-R on the cell surface. 

A follow-up experiment to immunostain LDL-R and qualitatively compare receptor density on A549, HepG2 and Vero cells, is shown in Appendix A. All cell lines contained LDL-R on the cell surface, with the highest surface density seen on HepG2 cells, and markedly lower, but similar density on A549 and Vero cells, indicating LDL-R density alone may not be a constraint on transfection in the A549 cells. Although further experiments would be needed to explain uptake and pharmacokinetic differences, cell lines that are difficult to transfect have limited dynamic range, and therefore limited ability to resolve LNP differences if used in a potency assay. 

For HepG2 cells, a full sigmoidal transfection response was observed without the addition of ApoE to the media; however, supplementation with ApoE resulted in a steeper slope and an improvement in assay sensitivity as indicated by a 20 percent decrease in the EC50, as shown in Figure 1G. As seen with HeLa and HEpG2 cells, providing ApoE in excess can increase the density of ApoE on the LNP surface, leading to an increase in LNP avidity, subsequently enhancing the LNP uptake rate at all concentrations, and resulting in steeper slopes. 

Consistent with the role of LDL-R in mediating the endocytosis of LDL and enabling cholesterol to enter body cells, it is expressed on the surface of many tissue types, and LDL-R has been measured in the transcriptome of many human-derived cell lines (The Human Protein Atlas), providing multiple potential candidate cell substrates for an LNP potency assay. A subset of those cell lines, including several examined here, have also been used for prior studies, demonstrating the endocytosis of LNPs designed for gene silencing and antigen expression. Although LNPs can readily enter different cell types, a quantitative comparison of RSV-F expression efficiency over a wide LNP concentration range showed that several commonly used model cell lines are not ideal substates for use in a potency assay. HepG2 cells provided a full sigmoidal dose–response curve, with upper asymptotes converging on complete cell transfection (>90%), enabling wide dynamic and dose response ranges, and therefore providing a suitable substrate for mRNA vaccine potency assays. 

Beyond selection of the most appropriate cell line based on mRNA uptake and ease of cell handling, several other factors play a key role in determining if the assay is sensitive to the most relevant process development parameters to be routinely tested to assess and aid in the rational design of mRNA vaccines. Here, we evaluate the HepG2 cells to interrogate the potency of mRNA encapsulated in LNPs with varying ratios of N/P (nitrogen to phosphate ratio in the lipid nanoparticles), varying LNP size and hydrodynamic diameter and varying concentrations of PEG. We show that the HepG2 cells are sensitive to changes in all of these parameters as reflected in the corresponding increase or decrease in the expression of the fraction of cells expressing the RSV-F protein. 

## Figures and Tables

**Figure 1 vaccines-11-01224-f001:**
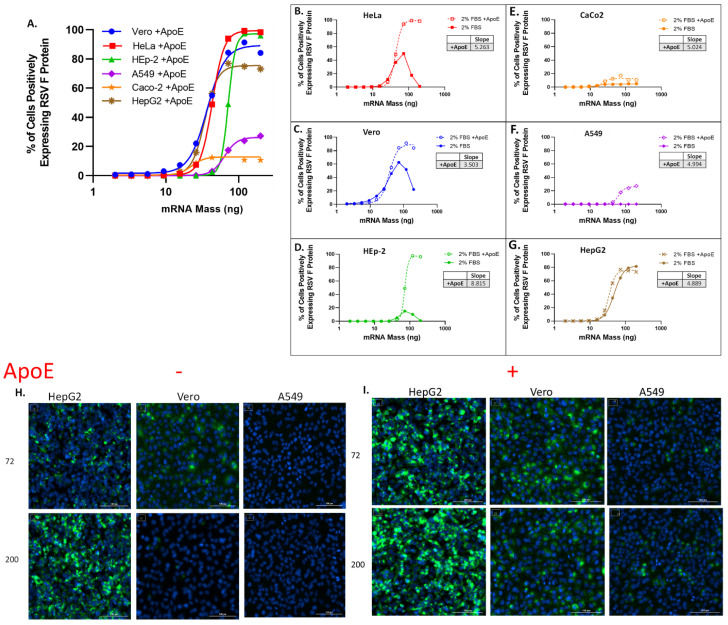
LNP transfection curves across varying cell lines indicate variable levels of transfection, which improve with the addition of ApoE during transfection. (**A**) RSV-F protein expression efficiency in Vero, HeLA, HEp-2, A549, CaCo-2 and HepG2 cells was measured by counting percentage of cells positive for RSV-F protein. All cells were transfected for 16 ± 2 h with titration of LNPs starting with 200 ng dose of mRNA in media with 2% FBS. Data were fit using variable slope-four parameter logistics regression (4-PL) model in GraphPad Prism software (version: 6.0). (**B**–**G**) RSV-F protein expression efficiency following transfection with LNPs in media + 2%FBS, and media + 2%FBS supplemented with ApoE at 4 ug/mL. Hill-slope values [Y = Bottom + (Top − Bottom)/(1 + 10^((LogEC50-X) × Hill-slope)] from 4-PL regression are given for curves obtained with FBS supplemented with ApoE. (**B**) HeLa cells. (**C**) Vero cells. (**D**) Hep-2 cells. (**E**) CaCo-2 cells. (**F**) A549 cells. (**G**) HepG2 cells. (**H**) Representative immunofluorescence images of HepG2, Vero and A549 cells without ApoE and (**I**) with ApoE shown at 72 (top row) and 200 ng/mL (bottom row) of mRNA dose representing the bottom and top of the dose response curve, respectively.

**Figure 2 vaccines-11-01224-f002:**
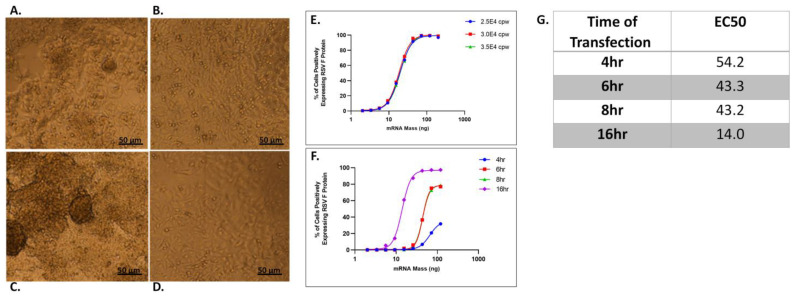
Optimization of cell plating parameters and transfection kinetics. Morphological characteristics of HepG2 monolayers on tissue culture plates and collagen I-coated plates at different cell seeding densities along with transfection kinetics of RSV-F protein encoding mRNA encapsulated within LNP. (**A**) HepG2 cells seeded at 7.5 × 10^4^ cells/well on tissue culture plates. (**B**) HepG2 cells seeded at 7.5 × 10^4^ cells/well on collagen I plates. (**C**) HepG2 cells seeded at 5.0 × 10^4^ cells/well on collagen I plates. (**D**) HepG2 cells seeded at 2.5 × 10^4^ cells/well on collagen I plates. (**E**) Comparison of RSV-F protein expression efficiency following transfection with LNPs at seeding densities of 2.5, 3.0 and 3.5 × 10^4^ cells/well. (**F**) RSV-F expression efficiency at 4, 6, 8 and 16 h post-transfection and (**G**) EC50 of dose–response is tabulated in ng of mRNA at each transfection time period.

**Figure 3 vaccines-11-01224-f003:**
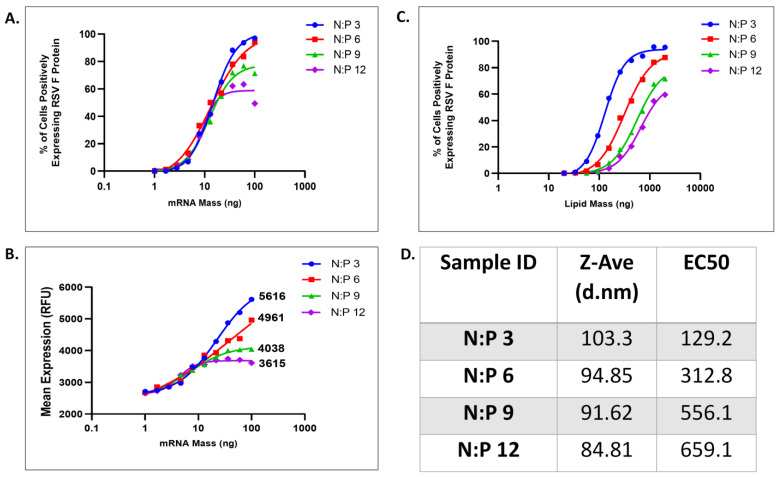
Potency assay is stability-indicating with the ability to discriminate changes in mRNA content in relation to LNP. (**A**) RSV-F expression efficiency for LNPs with N/P ratios of 3, 6, 9 and 12 with transfection and protein expression graphed as a function of mRNA mass. (**B**) RSV-F mean fluorescence per cell for LNPs with N/P ratios of 3, 6, 9, and 12 graphed as a function of mRNA mass. (**C**) RSV-F expression efficiency for LNPs with N/P ratios of 3, 6, 9, and 12 graphed as a function of lipid mass, to approximate particle number. (**D**) LNP particle size for differing N/P ratios and associated EC50 values for Figure 3C.

**Figure 4 vaccines-11-01224-f004:**
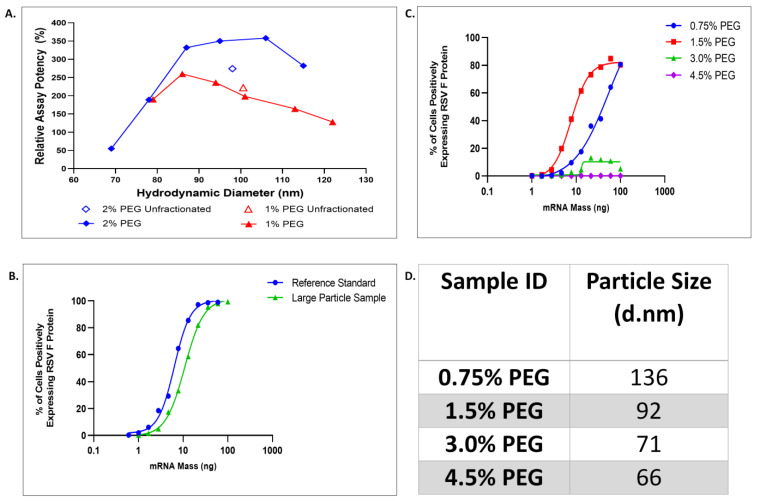
LNP transfection in HepG2 cells is optimal within a narrow size and PEGylation range. (**A**) Percent relative assay potency for 1% and 2% PEG-DMG unfractionated samples compared to samples that underwent density gradient centrifugation resulting in 6 differing size fractions. (**B**) RSV-F expression efficiency for sample lots with differing mean LNP size. (**C**) RSV-F expression efficiency for LNPs with PEG composition of 0.75, 1.5, 3.0 and 4.5% and (**D**) associated particle size.

**Table 1 vaccines-11-01224-t001:** Commercial adherent cell line properties.

Name	Species	Tissue	Cell Type	Growth Properties
HepG2	Human	Hepatocellular Carcinoma	Epithelial-Like	Adherent
A549	Human	Alveolar Carcinoma	Epithelial	Adherent
HeLa	Human	Cervix Adenocarcinoma	Epithelial-Like	Adherent
HEp-2	Human	HeLa Contaminant	Epithelial-Like (carcinoma)	Adherent
Caco-2	Human	Colorectal Adenocarcinoma	Epithelial-Like	Adherent
Vero	Monkey	Kidney	Epithelial	Adherent

## Data Availability

The data presented in this study are available on request from the corresponding author. The data are not publicly available due to the proprietary nature of the work which was conducted in compliance with requirements of the current legal framework at MSD. Data pseudo-anonymized are however available from the MRL labs upon reasonable request to any researcher wishing to use them for non-commercial purposes and will have to be approved by Merck legal prior to sharing.

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
