# Peer review of "Development and Characterization of an In Vitro Cell-Based Assay to Predict Potency of mRNA–LNP-Based Vaccines"

_vaccines, 2023, doi:10.3390/vaccines11071224_

Round 1

Reviewer 1 Report

Thank you for the opportunity to review the manuscript entitled “Development and characterization of a cell-based assay for mRNA-LNP vaccine potency”. An in vitro assay that can be used to predict vaccine performance or the immune responses in vivo is an important tool for vaccine development efforts and this study adds value in this direction. Please find below comments and suggestions.

Line 2:

The title would read better as “Development of an in vitro cell-based assay to predict potency of mRNA-LNP based vaccines”

 Line 21:

…. including cell-monolayer morphology, antigen protein expression kinetics, and assay….

Suggestion: Better to use just ‘antigen’ expression kinetics

…. including cell-monolayer morphology, antigen expression kinetics, and assay….

Line 24: ….

 the development of future mRNA vaccines.

Please amend as

 the development of future mRNA-based vaccines.

Line 38: …..

 ….progress made developing LNP delivery systems for the IV administration of small….

 Please amend as

 progress made after development of LNP delivery systems for the IV administration of small….

 Line 44: Please describe ‘DSPC’

 Line 71: ….reducing LNP uptake….

Please indicate ….reducing LNP uptake by what ?

Line 78: Please mention the size range.

Some of the sentences are too long to keep the focus of the statement, such as  

Line 90-94:

Lone 92: Describe ‘IM’ when mentioned for the first time in the manuscript.

Line 123:pCO2

Please correct CO2

Line 124:

Please describe full form of RSV, and describe why this antigen was selected.

Line 125: ARPE-19 cells are not described in the ‘Cell culture’ section under Materials and methods. Please clarify.

Line 140: Transfection was allowed at…..

Please clarify if the transfection was performed by only incubating the LNPs with the cells for 16 hours ?

CO2…please correct.

Line 147: ….followed with an anti-species, Goat Anti-Human, IgG Alexa Fluor® 488 fluoro

Please amend as

followed by Goat Anti-Human IgG Alexa Fluor® 488 fluorophore….

Or

followed by goat anti-species IgG Alexa Fluor® 488 fluorophore….

Line 176: ,,please clarify ‘plant medium’

Line 189: the word ‘then’ used twice.

Line 202: Please mention g value instead of rpm.

Line 203: After centrifugation, about 1 mL fractions were removed

What did the fraction contain?

Please mention …… After centrifugation, about 1 mL fractions containing………..? were removed

Line 330:

….reduction in EC50…

What does EC50 relate to here and what is its unit?

Line 486: LNPs with PEG composition of 0.75, 1.5, 3.0 and 4.5 mol percent

Correct as ….LNPs with PEG composition of 0.75, 1.5, 3.0 and 4.5%

Line 516: ….the sentence eds at ….other cell lines tested. 

Line 539: ….including several examined here, have been…

Please amend: including several examined here, have also been….

PEG and DMG full forms should be given when mentioned for first time in the manuscript.

Other relevant comments and suggestions.

1. The authors propose an in vitro assay that can be used as a tool to determine optimum biophysical conditions of mRNA/LNP vaccine particles based on expression levels of a candidate antigen. An in vitro assay that can be used to predict vaccine performance or the immune responses in vivo is an important tool for vaccine development efforts. The purpose of such an assay here is to down select optimum conditions of mRNA/LNP vaccine development process, and also to correlate these to optimum vaccine potency in vivo. The authors describe development and optimization of such a cell-based assay in the current manuscript. The final readout of the assay are expression levels of the vaccine antigen in cell lines. Thus, more appropriately this study describes the delivery efficacy of mRNA/LNP and quantitation of expression levels of RSV-F protein (as a candidate). In order to demonstrate that the assay readout translates to vaccine potency, it will be important to show correlation of antigen expression results under varying conditions tested (mRNA mass, LNP mass, PEG% etc) with induction and quality of the vaccine-elicited immune responses upon immunization of small animals (mice). This will substantially add value to the manuscript and the assay. Further it will be useful to validate the method or assay using more than one candidate antigen to generalize the usefulness of the methods described.

2. Please provide a list of abbreviated items mentioned in the manuscript, and their full forms at the end of the manuscript.

3. The references are not given as per MDPI Vaccines format. Please amend.

4. The supplementary file content does not appear to be the final version for submission. Please check.

Quality of English language is fine for submission. There are few sentences which need rephrasing. Also, there are few sentences that are very long, making the interpretation of the statement difficult. These can be also be rephrased (for example Lines 90-94; 493-498).  

Author Response

Please find our detailed response attached. Thank you very much for the extensive review of our manuscript.

Most appreciatively, 

Malini Mukherjee, PhD

Reviewer 2 Report

The article submitted to Vaccines journal by Nisarg Patel et al. entitled "Development and characterization of a cell-based assay for mRNA-LNP vaccine potency" is novel, well written and conclusions drawn were logical and the presented data supporting the conclusions.

Here I am accepting to publish the article in the Vaccines jornal with some corrections.  

1. why authors choose RSV-F as their Vaccine Candidate (some  write up about RSV-F will be great).

2.Figure 2 A-D can you provide clear microscopic images with higher magnification?

All the best

Author Response

Thank you very much for the positive and encouraging review of our manuscript. Please find the detailed response to your questions attached. 

Most appreciatively,

Malini

Reviewer 3 Report

Known in the field based on previous literatures:

1.  The use of nanoparticles and their carriers in drug delivery and treatment of various disease are well established and used globally.

2. With scientific development and nanoparticle technologies allow the accelerated development of messenger RNA (mRNA)-based vaccines within short period of time and currently two leading vaccines against COVID-19 based on it.

3. The success of the mRNA- based vaccines has served as an important validation of the safety and efficacy of this approach and drug developers are now using mRNA vaccines to address a broad array of new therapeutic areas.

4. The lipid nano particles (LNP) are used to chosen as a carrier vehicle to protect the mRNA from degradation and assist intracellular delivery and endosomal escape. The LNPs consist of a mixture of different lipids- cholesterol, PEGylated lipids, phospholipids, and cationic or ionizable lipids and each of them play important role.

In this manuscript authors reported following findings:

I have gone through the article titled ‘Development and characterization of a cell-based assay for mRNA-LNP vaccine potency’. Authors studied the antigen protein expression kinetics and assay sensitivity to LNP (PEG-lipid composition, mRNA mass, and LNP size). Authors performed and reported following findings-

1. Different cell line shows different properties hence their screening revealed significant differences in RSV-F protein expression

2. Authors optimized the cell plating parameters (collagen I coating at different cell seeding) and transfection kinetics

3. Examined and analyzed the PEG-lipid composition and LNP sizes to sensitivity of the assay.

The article presented are interesting and generally supportive of the conclusions drawn. There are, however, several issues which need authors attention. The following minor suggestions if incorporated could help in the better understanding of the significance of the work and implications.

Minor Concerns:

1. Authors should write and introduce about RSV- F protein in introduction or where they use 1st time.

2. It’s hard to read the axis in figure 1. Please increase the font size of figure 1 (B, C, D, E, F and G).

3. In figure 3 (D)- there is no significant difference of particle size of samples- N:P 6 and N:P 9, but there is obvious change in EC50. Explain the reason and discuss it.  

4. There are many articles available related to LNP. Explain, how your article is different from rest how does it address a specific gap in the field?

Author Response

Thank you very much for the positive and encouraging review of our manuscript. Please find a detailed response to your questions attached. 

Most appreciatively,

Malini Mukherjee, PhD
